# Let's Think Dot by Dot:
# Hidden Computation in Transformer Language Models

**Jacob Pfau, William Merrill & Samuel R. Bowman**
Center for Data Science
New York University
NY, NY 10012, USA
{jp6263,willm,bowman}@nyu.edu

## Abstract

Chain-of-thought responses from language models improve performance across most benchmarks. However, it remains unclear to what extent these performance gains can be attributed to human-like task decomposition or simply the greater computation that additional tokens allow. We show that transformers can use meaningless filler tokens (e.g., '......') in place of a chain of thought to solve two hard algorithmic tasks they could not solve when responding without intermediate tokens. However, we find empirically that *learning* to use filler tokens is difficult and requires specific, dense supervision to converge. We also provide a theoretical conjecture for the class of problems where filler tokens are useful in terms of the *quantifier depth* of a first-order formula. For problems satisfying this characterization, chain-of-thought tokens need not provide information about the intermediate computational steps involved in multi-token computations. In summary, our results show that additional tokens can provide computational benefits independent of token choice. The fact that intermediate tokens can act as filler tokens raises concerns about large language models engaging in unauditable, hidden computations that are increasingly detached from the observed chain-of-thought tokens.[1]

## 1 Introduction

Chain-of-thought reasoning improves language model (LM) performance when compared to direct, no chain-of-thought, responses (Wei et al., 2023; Suzgun et al., 2022; Lanham et al., 2023). However, recent empirical work shows that answers arrived at via chains of thought frequently are not faithful to the intermediate reasoning steps taken within the chain (Lanham et al., 2023; Turpin et al., 2023). As a limit case of unfaithfulness, the *filler token* setting replaces chain-of-thought tokens with arbitrary, repeated tokens, e.g. '......', as shown in Figure 1. By comparing language model performance when given filler tokens instead of chains of thought, we can assess whether a given LM is capable of carrying out cross-token computations that are not reflected in the chain of thought tokens.

The most widely used LM alignment methods are purely behavioral. Reinforcement learning from human feedback, constitutional AI, instruction fine-tuning, and automated red-teaming all rely on judging or comparing model output tokens. LMs capable of making use of filler tokens undermine this reliance because the reasoning carried out across filler tokens cannot be judged from the tokens themselves.

In this work, we study the strict filler case where filler tokens are repeated dots, '......'; however, the utility of such tokens depends only on the availability of excess capacity in activation space. The '......' case is a minimal version of the more general setting where any sequence of filler tokens is provided between an input prompt and some complex output token. For example, the filler sequence could be "Lorem ipsum dolor sit amet, ..." or

---

[1]Code is available at https://github.com/JacobPfau/fillerTokens

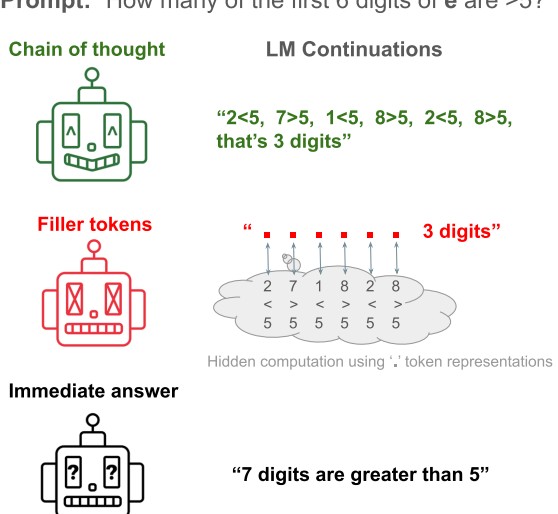

Figure 1: A cartoon contrasting three LM question-answering protocols: chain of thought, filler tokens, and immediate answer. In the filler-tokens setting, the LM uses arbitrary, irrelevant intermediate tokens (e.g., '......') before answering, but the filler tokens' hidden-layer representations still provide computation relevant to later tokens. In previous work, chain-of-thought was shown to allow greater expressive power than immediate answer (correspondingly, the 'immediate answer' bot gives the wrong answer). We show that filler tokens can, on certain tasks, match the performance of chain-of-thought reasoning.

repeating a question back to the user, as long as the string requires minimal computation and precedes a more algorithmically demanding token.

Empirically, commercial large language models (LLMs) do not benefit from filler tokens on common QA and math benchmarks; Claude 2 and GPT-3.5 achieve the same performance with filler tokens as they do when responding directly without intermediate tokens (Sachan, 2023; Lanham et al., 2023). However, current LLMs' limitations cannot be extrapolated to larger scales: The empirical evidence on current LLMs does not clarify whether a failure to use filler tokens is an in-principle limitation of transformer expressivity (or their loss landscapes), or instead, if filler token performance may arise at larger scale. Additionally, it is unclear whether these evaluations targeted tasks where filler tokens would be beneficial. In this work, we demonstrate that transformers trained on the next-token prediction objective *can achieve improved performance on certain tasks when given filler tokens*, achieving perfect accuracy whereas the no-filler, immediate-answer setting achieves only low accuracy.

These results also provide interesting insight into how filler tokens extend the expressive power of transformers. As single-token predictors, transformers can only solve problems in a complexity class called $\mathsf{TC}^0$, which means transformers cannot express problems like permutation composition or graph connectivity (Merrill & Sabharwal, 2023a; Strobl et al., 2023). Whereas linear or polynomial chain-of-thought steps can add power to transformers beyond $\mathsf{TC}^0$ (Merrill & Sabharwal, 2023a), transformers remain in $\mathsf{TC}^0$ with even a polynomial number of filler tokens. Thus, unlike for chain of thought, we cannot expect filler tokens to let transformers solve problems outside $\mathsf{TC}^0$, e.g. graph connectivity. However, our results suggest that *filler tokens likely extend the expressive power of transformers within* $\mathsf{TC}^0$. In particular, our results establish that reasoning requiring many *nested quantifiers* becomes expressible for transformers with filler tokens whereas it is conjectured that no-intermediate-token, immediate-answer transformers cannot solve these problems. We propose synthetic tasks for which transformers without chain of thought have been conjectured inadequate in expressivity (Sanford et al., 2024) and show that *using filler tokens, transformers can solve these tasks.*

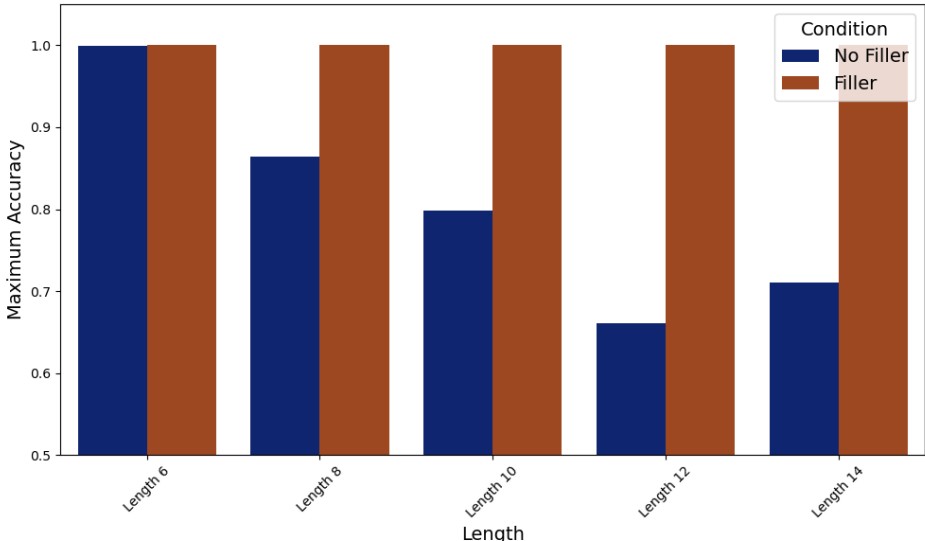

Figure 2: The performance gap between a transformer, Llama 34M, with and without filler tokens increases with 3SUM problem length up to length 12, showing that filler tokens reliably provide an advantage for sufficiently complex 3SUM problems. The immediate-answer models were trained for $5\times$ the number of steps.

Our contributions are the following:

1. We construct two synthetic datasets, 3SUM (Figure 3) and 2SUM-Transform, on which LLAMA transformers fail to solve the task without filler, but achieve 100% and 94% accuracy, respectively, when provided filler tokens.

2. We find that filler token performance increases over immediate answers as the length and complexity of inputs increase (Figures 2 and 4).

3. We contextualize filler tokens with respect to theoretical expressivity results highlighting that filler-token prompting remains within circuit complexity class $\mathsf{TC}^0$, but we provide empirical evidence that filler-tokens may add power *within* $\mathsf{TC}^0$.

4. We find that learning to use filler tokens is difficult and requires specific, dense supervision to converge. Standard chain-of-thought data is insufficient for models to learn to leverage filler tokens effectively, c.f. Section 4.3.

Taken together these findings suggest that although current LLMs are unlikely to benefit from filler tokens, this is not an in-principle limitation of current architectures. Given demonstrations of parallelizable task decompositions, we expect that current LLMs would also realize benefits from filler tokens.

## 2 Background and Related Work

### 2.1 Transformer Expressivity and Filler Tokens

Recent theoretical work establishes that transformers without additional reasoning tokens are limited to solving only highly parallelizable problems (see Strobl et al., 2023 for an overview). Formally, Merrill & Sabharwal (2023a) place log-precision transformers in the circuit complexity class $\mathsf{TC}^0$, which can be equivalently understood as the class of problems definable in first-order logic with majority quantifiers (Merrill & Sabharwal, 2023b). It follows that problems outside $\mathsf{TC}^0$ (those that cannot be defined in first-order majority logic) *cannot* be solved by transformers without additional reasoning tokens. This

includes canonical reasoning problems like composing permutations, graph connectivity, or evaluating boolean formulas. This suggests that—without additional reasoning tokens—transformers are surprisingly limited.

A natural way to get around these expressiveness limitations is to provide the transformer additional reasoning tokens. When transformers have a chain of thought (i.e., can generate tokens that get added to their input), they can indeed solve problems outside $\mathsf{TC}^0$ if the chain of thought is long enough (Merrill & Sabharwal, 2023c; Feng et al., 2023). These results show that chain of thought, in addition to providing a particular decomposition hint for a complex problem, expands the computational power of transformers in a way that is essential for many types of sequential reasoning problems.

**Filler tokens and expressivity** In the filler tokens setting, i.e., when the context is expanded by appending blank tokens, the model clearly cannot benefit from having instructions to follow, but is there still a computational benefit? As long as the number of filler tokens is polynomial, the argument of Merrill & Sabharwal (2023a) goes through to show that transformers with filler tokens can only solve problems in $\mathsf{TC}^0$. Merrill & Sabharwal (2023a) show for inputs of size $n$, a transformer can be simulated by an $O(1)$ depth, $poly(n)$ size threshold circuit. If we add polynomial filler tokens, this implies we can simulate the circuit with $O(1)$ depth and $poly(poly(n)) = poly(n)$ size. As a result, filler tokens can only increase expressivity of transformers *within* $\mathsf{TC}^0$.

To understand the expressivity gains from filler tokens, we can look for classes of problems *within* $\mathsf{TC}^0$ that transformers without filler tokens cannot express. Merrill & Sabharwal (2023b) conjecture that there are problems in $\mathsf{TC}^0$ that fundamentally require resolving many quantifiers at the same time, which therefore cannot be expressed by transformers without filler tokens.[2]

Filler tokens provide a natural construction for solving problems with deep quantifier nesting, given access to appropriate positional encodings:[3] a problem requiring quantifier depth $k$ can be expressed with $n^k$ filler tokens by using the filler tokens to enumerate over quantified values. However, no complexity theoretic hierarchy has been proven within $\mathsf{TC}^0$, so absent such a result, we provide empirical evidence that certain nested quantifier problems cannot be solved without filler tokens, while transformers can learn to solve them perfectly with filler tokens. *The conjectured expressivity picture compatible with our empirical results is: (1) vanilla, constant length input transformers can solve a strict subset of $\mathsf{TC}^0$ problems; (2) transformers with polynomially many filler tokens can solve a greater subset of $\mathsf{TC}^0$ problems (3) transformers with sufficiently long chain-of-thought can solve a strict superset of $\mathsf{TC}^0$ problems.*

## 2.2 Further related work

**Empirical Results on Non-myopic Computation in Transformers** Lanham et al. (2023) and Sachan (2023) both find that, for commercial LLMs, filler tokens generically fail to improve performance over immediate answers when evaluated on NLP and mathematics QA benchmarks.

Previous and concurrent research identified cases where token representations contribute to the prediction of tokens occurring multiple indices later showing that, in practice, such contributions both reduce loss on the average case (Janus, 2023; Wu et al., 2024) and can

---

[2]There is a close connection between the quantifier depth of a formula and the size (width) of a $\mathsf{TC}^0$ circuit for simulating it. The conjecture of Merrill & Sabharwal (2023b) is essentially the assumption that there is a *size hierarchy* within $\mathsf{TC}^0$: larger circuits are more expressive. While intuitive, no size hierarchy theorem has been proven for $\mathsf{TC}^0$ itself, though such theorems exist for related circuit classes like $\mathsf{AC}^0$ (cf. Limaye et al., 2019). From this point of view, one way to understand the benefit of filler tokens is that they increase the width of the transformer computation graph (but not its depth, which standard CoT would increase).

[3]In particular, the construction requires computing mod with position arguments. With standard positional encodings, it is not clear whether it is possible to express mod in general over all positions.

be mechanistically identified via probing (Pal et al., 2023). Complementing these works, we propose filler tokens as a limit case for of coordinated, token-agnostic, non-myopic computation; this case is of particular interest for its expressivity and alignment properties.

**Transformer Variants Using Adaptive Computation**   Recent work has also proposed training transformers to predict when further computation is needed for token predictions using pause tokens (Goyal et al., 2024) or meta-tokens (Zelikman et al., 2024). Whereas Goyal et al. (2024) and Zelikman et al. (2024) address the *engineering* question of how to modify the transformer architecture, language modeling objective, and tokenization process to allow adaptive, filler-like computation; our work addresses the *scientific* question of under what conditions standard, causal transformers, on the unmodified next-token prediction task, can learn to use intermediate tokens as filler tokens.

## 3   Synthetic data: 3SUM and 2SUM

We would like to understand why previous results found *no* performance increase from filler tokens on tested LLMs (Lanham et al., 2023). By finding synthetic tasks on which filler tokens improve LM performance, we can determine (1) what kinds of evaluation data can benefit from filler tokens, and (2) what kinds of training data are needed to teach models to use filler tokens (c.f. Section 4.3). To answer these questions, we construct two synthetic datasets each highlighting a distinct condition under which filler tokens provide performance improvement to transformers.

**3SUM**   The motivation for this problem comes from two directions. *Theoretically*, 3SUM is of interest since it is likely not expressible with a single forward pass (as it has quantifier depth greater than 2; c.f. Equation (1)) but is parallelizable–therefore amenable to filler tokens. *Intuitively*, 3SUM involves simply matching triples of in-context inputs by their meaning. So a demonstration that 3SUM is learnable using filler tokens provides evidence of an expressivity gap between the filler and immediate-answer setting for the general class of nested quantifier resolution problems.

**2SUM-Transform**   A secondary, simpler task: Match pairs of inputs (summing to zero), where input tokens are obfuscated by a transformation only specified in the *final* token of the input sequence. Leaving the input under-defined until this final transform token prevents in-place computation over input tokens' forward passes. The 2SUM-Transform problem is an instance of the more general format in which a question is posed at the end of a long input, as when presenting a document followed by a question about the document.

### 3.1   3SUM Definition and Tokenization

Figure 3 diagrams a simple example of the 3SUM[4] problem and the accompanying chain of thought. Formally the 3SUM task statement is: Given $[x_0, \ldots, x_n]$, $x_i \in \mathbb{Z}_{10}^d$ as input, predict whether the statement

$$\exists x_i, \exists x_j, \exists x_k : x_i + x_j + x_k = \mathbf{0} \mod 10 \tag{1}$$

is true. Indices i,j,k must be distinct.

In the worst case, this task requires considering $C_3^n$ summations, i.e. $O(n^3)$ operations. A standard layer of attention induces only quadratic dependencies between following layer activations and previous layer inputs. Hence, heuristically the 3SUM problem naturally exceeds the expressive capacity of a transformer for large $n$.[5]

---

[4]Sanford et al. (2024) name a variant of this problem 'Match-3'. In the Sanford variant, the inputs $x_i$ and predictions are both multi-hot vectors.

[5]The limits of a given transformer size's expressivity (bounding parameter-counts performance by 3SUM length) is heuristically calculated in Sanford et al. (2024) but the bound is unrealistically large given learning constraints–suggesting small transformers of 10M parameters can solve 3SUM for lengths up to 10,000 inputs. This is a loose bound, which our results show is unrealistic, and a realistic analysis could use sensitivity bounds on learnability as in e.g. Hahn & Rofin (2024).

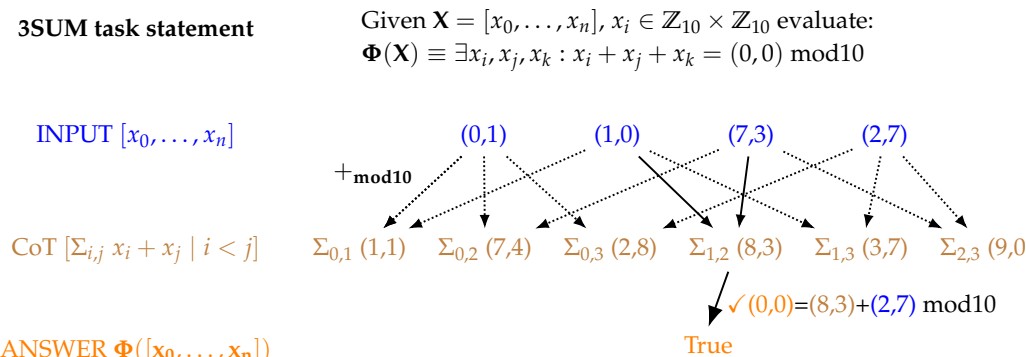

**3SUM task statement**

Given $\mathbf{X} = [x_0, \ldots, x_n]$, $x_i \in \mathbb{Z}_{10} \times \mathbb{Z}_{10}$ evaluate:
$\mathbf{\Phi}(\mathbf{X}) \equiv \exists x_i, x_j, x_k : x_i + x_j + x_k = (0,0) \bmod 10$

INPUT $[x_0, \ldots, x_n]$    (0,1)   (1,0)   (7,3)   (2,7)

$+_{\mathbf{mod10}}$

CoT $[\Sigma_{i,j}\, x_i + x_j \mid i < j]$   $\Sigma_{0,1}$ (1,1)   $\Sigma_{0,2}$ (7,4)   $\Sigma_{0,3}$ (2,8)   $\Sigma_{1,2}$ (8,3)   $\Sigma_{1,3}$ (3,7)   $\Sigma_{2,3}$ (9,0)

$\checkmark$ (0,0)=(8,3)+(2,7) mod10

ANSWER $\mathbf{\Phi}([\mathbf{x_0}, \ldots, \mathbf{x_n}])$    True

Figure 3: 3SUM involves finding matching triples that sum to the zero vector modulo 10. The chain-of-thought row demonstrates a decomposition of the 3SUM problem into parallelizable, pairwise summations, which can be calculated using filler tokens. All pairs are summed in lexicographic order by index. The resulting sequence is "01 10 73 27 : 11 74 28 83 37 90 ANS True". We train on a mixture of such chain-of-thought sequences and filler sequences which replace the chain-of-thought tokens with '.'s. In practice, we add additional positional encoding information to the input and chain-of-thought sequence to simplify the task, as described in Section 3.1. In the general case we vary input sequence length and tuple-dimensionality to study the effects of data complexity on filler tokens.

Our sequence data consists of *input* e.g. "A01 B10 C73 D27", *intermediate tokens* e.g. ". . .", and *3SUM-label* e.g. "True". Here, "A05" denotes the tuple $(0,5)$ and $A$ marks this as the first input, $x_0$. Inputs are vectorized as multi-hot binary vectors passed to the model as embedding vectors followed by a learned linear layer. The input vectors have masked labels and so do not contribute to the loss.[6]

We consider three different types of intermediate-token sequences to insert between the problem input and output:

1. **Filler** These sequences use ". . .", repeated dots, as intermediate tokens e.g. "A05 B75 C22 D13 : . . . . . . . . . . . . ANS True". These tokens correspond one-to-one with the chain-of-thought tokens below. Each dot is a separate token for a total of $n^2$ intermediate tokens.

2. **Chain of Thought (Parallelizable CoT Solution)** These sequences are of the form: "A05 B75 C22 D13 : AB 70 AC 27 AD 18 BC 97 BD 88 CD B ANS True".[7] This chain of thought reduces the 3SUM problem to a sequence of 2SUM problems by writing all relevant intermediate summations (as shown in Figure 3). These pairwise sums reduce the cubic cost of 3SUM to the quadratic cost of checking whether an input $x_i$ exists which matches each pairwise sum–this check can be done using just one attention layer. For each intermediate 2SUM result, if that result matches a third input, we write the index of the third input instead of the sum, as seen at the end of

---

[6]Besides the masked input tokens, all subsequent tokens are presented as one-hot labels so as to be compatible with the standard cross-entropy language modeling objective. This choice of inputs as embedding vectors and the rest as one-hot tokens is admittedly non-standard (though Sanford et al. (2024) do the same), and was made to reduce the scale of compute needed to realize the separation between immediate-answer and filler settings. To realize the same filler-token to immediate-answer compute gap when using one-hot, digit-wise tokenization of inputs, we would have to increase input length by $n_{new} = n\sqrt{2d}$ which would 2-4x compute cost.

[7]In practice, we reduce the vocabulary size to accelerate training. We randomly drop one of each paired character in the chain of thought yielding e.g. "A05 B75 C22 D13 : A 7 C 2 D 1 B 9 D 8 C B ANS True". This change is superficial, since to achieve optimal loss, the predictor must still predict the tokens equivalent to the original (spreading probability mass uniformly). Since wall-clock time for individual gradient steps is linear in sequence length, this change saves us up to a factor of $d$, input dimension, in wall-clock time.

the chain with "CD B". We choose this particular task decomposition for the chain of thought because it is fully parallelizable.

3. **Chain of Thought (Instance-Adaptive CoT Solution)** These sequences are of the form: "A15 B75 C22 D13 : A B C 15 75 22 2 B C D 75 22 13 0 ANS True" (the data-generating process is described in Appendix C). In the previous, parallelizable solution we neatly factored 3SUM token-wise into parallelizable sub-problems using the same uniform decomposition across all problem instances. However, human reasoning, and the resulting chains of thought, are flexible using instance-specific heuristics to decompose problems as best suits the problem at hand. In particular, when the computation carried out in a later chain-of-thought token depends on the results found in an earlier chain-of-thought token we term this *instance-adaptive computation*. This kind of computation is incompatible with the parallel structure of filler token computation. Consequently, in order to use filler tokens on natural language data, LLMs would need to *discover* parallelizable algorithmic solutions given access only to CoT demonstrations lacking parallel structure. By training on instance-adaptive chains of thought, we can study whether models can learn to use filler tokens having seen only more naturalistic chain-of-thought data. These instance-adaptive chains of thought reduce the $d$-dimensional 3SUM problem to a sequence of one-dimensional 3SUM problems. Each triple which sums to zero in the first coordinate is evaluated in its other dimensions individually, so the worst-case[8] length for these chains of thought is $O(n^3)$. These serial chains of thought require caching of intermediate results (dimension-wise 3SUMs) and as such cannot be parallelized across filler tokens.

The parallelizable CoT solution provides supervision for an algorithm which can be implemented using filler-tokens. To implement this algorithm using filler tokens, individual '.' tokens compute individual 2SUM results by attending to pairs of inputs–this can be done in one layer. Then the following layer again attends over all inputs checking whether a third matching input exists. The final prediction token can then attend across hidden filler token representations to check whether there exists a representation encoding the zero vector, outputting 'True' if, and only if, 3SUM was satisfied.

## 3.2 2SUM-Transform

Formally the 2SUM problem is: Given $[x_0, \ldots, x_n]$, $x_i \in \mathbb{Z}_{10}^d$ as input, predict

$$N_{\text{sum}} = |\{x_i, x_j : x_i + x_j = 0 \bmod 10\}| \tag{2}$$

This can be done in a single forward pass with a standard transformer, so to demonstrate the utility of filler tokens, we propose the 2SUM-Transform problem in which a permutation[9] $P_k \in \mathbb{Z}_{10}^{d*n}$ is used to obscure the input sequence. This permutation shifts every digit of the input tokens by a random offset. The resulting 2SUM-Transform input is then $[P_k(x_0), \ldots, P_k(x_n), k]$. We randomly sample 10 such permutations[10] $\{P_0 \ldots P_9\}$ and uniformly at random sample a permutation to apply for each sample in the dataset.

For 2SUM, we use only linearly many chain-of-thought tokens which correspond to the un-transformed $[x_0, \ldots, x_n]$. Mimicking the realistic setting in which a LLM might use repeating back the question as filler tokens, we use filler token sequences of the following form for 2SUM: "97 80 94 44 $P_8$ 97 . 80 . 94 . ANS:4". Here, "97 80 94 44" are the permuted inputs $P_8(x_i)$; "$P_8$" denotes which permutation was applied; and "97 . 80 . 94 ." are the filler tokens–the input repeated back. We train on uniform mixtures of filler-token and chain-of-thought sequences. Chain-of-thought sequences are of the form: "17 84 09 39 $P_5$ 17 08 84 73 09 35 ANS:2", sequentially listing $P_k(x_i)$ and $x_i$.

---

[8]Given our compute constraints, we remove from the training set all sequences having length over 95th percentile.

[9]$\mathbb{Z}_j^i$ here denotes the $i$-fold direct product of the cyclic group on $n$ elements–i.e. each digit of every tuple is permuted independently.

[10]For 2SUM, a brute-force solution, without knowledge of the transform, requires $KN^2$ comparisons where $K$ is the number of possible transformations.

## 4   3SUM: transformers converge with filler tokens and fail without

### 4.1   Experimental Setup

We use a 34M-parameter Llama model and train with standard Adam hyper-parameters. We train on 10,000,000 samples and test on 2,000 samples. We train to convergence: for 5 epochs in the filler and chain-of-thought settings, and 25 epochs in the immediate-answer setting (see Appendix E for loss plots). We always report the per-run maximum validation performance across epochs, i.e. early-stop performance. Additional training and model details are included in Appendix A.

### 4.2   Results

To show that filler tokens confer greater expressive capacity, letting transformers solve hard problems, we must show that transformers *without* filler cannot solve our task, 3SUM. In particular, we require evidence that the non-learnability of 3SUM in the immediate-answer setting is due to expressive capacity and not simply a difference in the particularities of the data distribution and presentation. To this end, we show that for short enough inputs, 3SUM can be solved without filler tokens, but for longer inputs, 3SUM cannot be solved without filler tokens. Models converge to 100% accuracy on chain-of-thought data as well, but we do not show this in figures for simplicity. The length and dimension scaling experiments below were trained on a 50/50 split of filler and chain-of-thought data. In the below experiments, Figure 2 and Figure 4, we consider two cases:

1.  (Blue bars, Immediate-answer Case) **Test** on immediate-answer, no-intermediate-tokens, data. In Figure 2, **train** on immediate-answer, no-intermediate-tokens, data only. In Figure 4, train on a 50/50 mixture of CoT and immediate-answer data.[11]

2.  (Brown bars, Filler-tokens Case) **Test** on filler-token sequences only. **Train** on a uniform, 50/50, mixture of chain-of-thought and filler-token sequences.

**Length Scaling Transformers Consistently Benefit From Filler on Sufficiently Complex Inputs.**   Figure 2 shows that, as expected, for length-6, dimension-3 3SUM instances, 3SUM is learnable both with and without filler tokens. However, as we scale the length of inputs up to length 12, we find increasing performance gaps: The immediate-answer models achieve near-random accuracy at 66%, whereas with filler tokens, accuracy remains 100%.

**Filler Token Representations Encode Hidden, Task-Relevant Computation.**   To validate that the additional forward passes available when using filler tokens are used to compute sub-problems for the 3SUM task, we train a final-layer probe to decode intermediate results encoded at varying filler token positions. This experiment confirms that filler-token models use filler as an additional computational affordance not available to immediate-answer models; a priori, it might have been the case that filler-token models still do all computation in-place on input tokens, and merely have improved learning dynamics–probing rules out this latter explanation. Given a model trained on filler tokens, we fine-tune the final attention layer (freezing all earlier layers) to predict the solution given reduced numbers of filler tokens. Figure 5 shows that when decoding from learned representations on filler tokens, a frozen model yields monotonically improving predictions as we allow additional filler tokens. Additional ablation and probing details are described in Appendix B.

**Dimension Scaling Shows Filler Token Benefits at Shorter Sequence Lengths**   Figure 2 showed that to take advantage of filler tokens to solve more complex problems than immediate-answer response can, 4 layer models realize maximal benefit starting at length-12 3SUM inputs, that is sequences of token length $> 150$. Our experiments required $O(n^2)$ filler tokens to realize an expressivity gap over the immediate-answer response. This raises the question of whether LLMs with tens or hundreds of layers require prohibitively many

---

[11]Training data choice differs since we optimized it for baseline, immediate-answer performance.

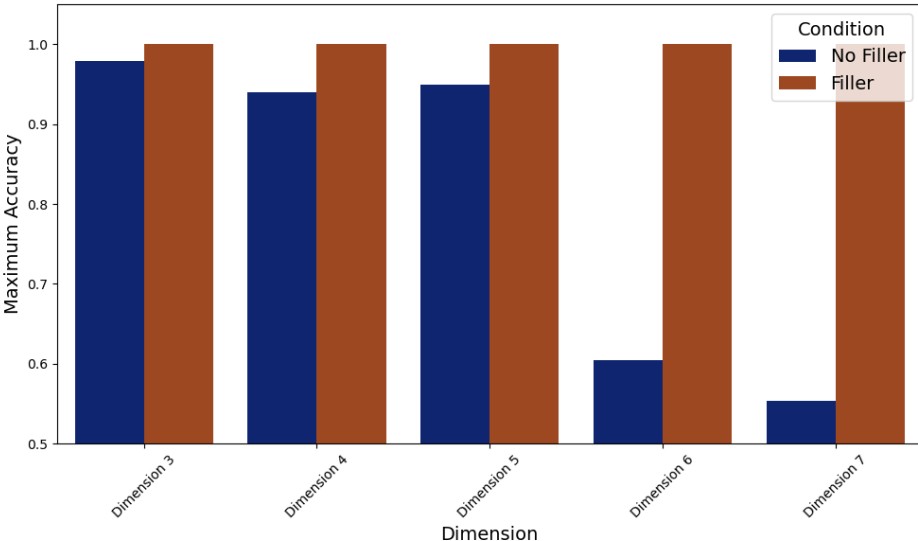

Figure 4: The filler-token to immediate-answer performance gap as a function of 3SUM tuple dimension. 3SUM input sequence length is fixed at length 8. The number of filler tokens is fixed across all runs, since we determine the number of filler tokens as a function of sequence length, not sequence dimension. This shows that LMs can realize performance increases using limited numbers of filler tokens on sufficiently complex data.

filler tokens to see improved performance over the immediate-answer baseline? To answer this question, we show the effects of scaling input *complexity*, i.e. 3SUM dimension, instead of input *length*, realizing performance gaps at lower filler token counts.

Figure 4 shows that for fixed length inputs, by increasing input dimension, we can realize performance gaps between the filler and the immediate-answer settings for even length-8 inputs. In these experiments, we used as our immediate-answer baseline a model trained on a 50/50 mixture of filler token sequences, and instance-adaptive CoT (the evaluation is done on the filler token sequences only). We use this mixed-dataset baseline rather than training on only immediate-response sequences, because the mixed-dataset models outperform the immediate-response-only models. In Appendix D, we provide further results on the effects of scaling tuple dimensionality for length-10 inputs.

### 4.3 Filler Tokens Only Improve Performance Given Parallelizable CoT Demonstrations

Despite transformers having the *expressive capacity* to solve certain filler-token tasks, learning filler token computations poses a hard *learning* problem. There are two reasons for this: First, it is impossible to densely supervise filler token solutions, because by assumption, filler tokens are used in precisely those cases when underlying, hidden computation decorrelates from the meaning of the corresponding tokens. Second, algorithms learned from chain-of-thought data generically require instance-adaptive, serial computation (Merrill & Sabharwal, 2023c)–such computation is incompatible with the parallel structure of filler-token compute.

To quantify the effects of these learning obstacles, we run two ablations: First, we train models on filler-token-only sequences to evaluate the difficulty of learning filler-token computation in the absence of parallelizable chain-of-thought data. In this case, we train on length-14, dimension-3 data, and performance remains at $\sim 71\%$ accuracy across all three random initializations. This performance is the same as the no-filler, immediate-answer condition observed in Figure 2.

In our second ablation, we train on a data mixture using instance-adaptive chain-of-thought sequences and filler tokens (described in Section 3.1). We find that **models trained on instance-adaptive CoT data fail to use filler tokens**. On filler token sequences, the resulting

| Data | Accuracy |
|---|---|
| Chain of Thought | **95.1%** |
| Filler | 93.6% |
| Immediate Answer | 78.7% |
| Majority Class Baseline | 63% |

Table 1: 2SUM: Highest observed performance across 5 runs per data type. The majority class baseline is above random accuracy because the data-generating process yield class imbalance. Models reach 75% accuracy within the first epoch. The immediate-answer condition narrowly improves over this 75% baseline, indicating that the immediate-answer model fails to learn significant algorithmic structure beyond label statistics.

models remain at, or below, immediate-answer, baseline performance, Figure 6. This indicates that there is *no* transfer from serial, instance-adaptive demonstrations to filler tokens for the 3SUM problem.

## 5 2SUM Experiments

In the 2SUM setting, a transformer with immediate answer performs well above random, but significantly below the same model when trained with filler, as shown in Table 1. Table 1 reports the maximum performance across five random initializations, because we observe significant variance across runs. The chain-of-thought and filler-token results use the same model but evaluate on disjoint subsets of the test data. Filler-token performance approaches chain-of-thought performance, recovering 90% of the benefits of chain-of-thought tokens over the immediate-answer baseline. We also experimented with training the immediate-answer model on a mixture of immediate-answer and chain-of-thought data; this under-performs relative to direct training on immediate-answer sequences. We use the same hyper-parameters as are used for 3SUM, c.f. Appendix A.

## 6 Conclusion

When are the benefits of chain-of-thought reasoning in transformer LMs due to interpretable, serial problem decompositions, or simply additional forward passes? We have seen that, for certain parallelizable problems, transformers achieve improved performance when given filler tokens instead of chain-of-thought tokens. This performance gap demonstrates that, given adequate training data, intermediate tokens between input and answer may be used purely for their computational capacity rather than for the human-like, faithful serial reasoning which such human-generated text represents. In such cases, the intermediate tokens are at best non-informative, as in the '......' case, and at worst misleading insofar as they describe reasoning unrelated to the computations occurring in intermediate-token, hidden representations.

We have offered a theoretical case for filler token usefulness, the quantifier depth $> 2$ case, and empirical evidence that filler token usage can be efficiently learned. Returning to our original question of whether LLMs should be expected to make use of filler tokens in the future, we can reduce the problem to asking: First, to what extent do token-parallelizable, $TC^0$ algorithmic problems arise in the natural language context? Second, to what extent does natural-language text provide adequate supervision for filler-token computation, providing parallelizable supervision rather than non-parallelizable, instance-adaptive chains of thought? If these conditions are met, we expect filler token usage to emerge in LLMs.

**Acknowledgments**

This project has benefited from financial support to Sam Bowman by Eric and Wendy Schmidt (made by recommendation of the Schmidt Futures program) and Open Philanthropy, and from in-kind support by the NYU High-Performance Computing Center. This

material is based upon work supported by the National Science Foundation under Grant No. 1922658. Any opinions, findings, and conclusions or recommendations expressed in this material are those of the author(s) and do not necessarily reflect the views of the National Science Foundation.

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

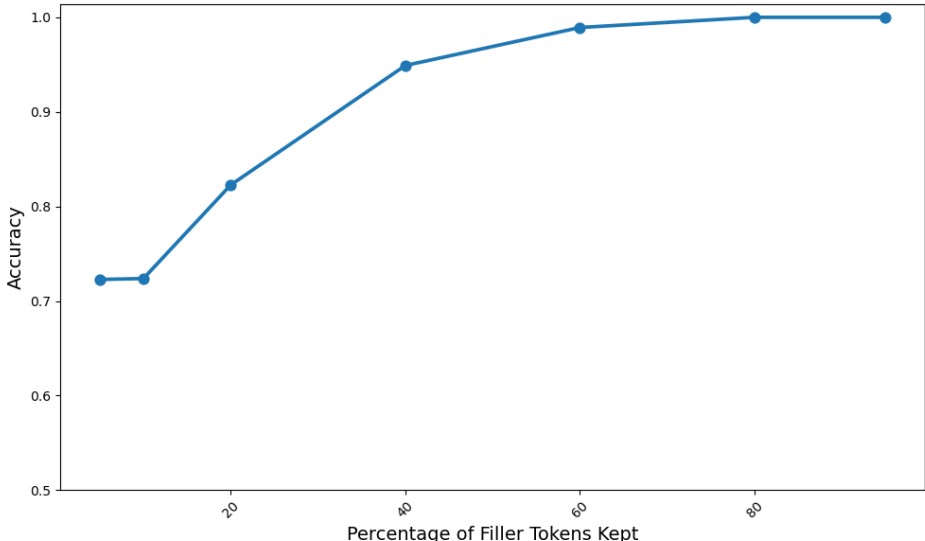

Figure 5: To verify that filler tokens are being used for hidden computation relevant to the final 3SUM prediction, we freeze model weights and finetune only the final attention layer to predict the 3SUM task given reduced numbers of filler tokens. Accuracy improves given access to additional filler tokens, suggesting that filler token representations encode hidden computation relevant to the 3SUM prediction task. The model was trained on length-14, dimension-3 3SUM instances, and originally trained on sequences having 184 filler tokens.

## A  Training Hyper-Parameters

The 34m-Llama model uses 4 layers, 384 hidden dimension, and 6 attention heads (Touvron et al., 2023). This is a scaled-down, randomly-initialized version of the Llama model. Input 2SUM and 3SUM vectors are given as hard-coded, multi-hot embedding vectors which are projected through a learned $(d_{input}, 384)$ dimensional linear layer.[12] Intermediate tokens (filler and chain-of-thought) are given as one-hot tokens. We use Adam with a learning rate of 1e-4 throughout. For all filler and chain-of-thought runs we use a weight decay of 0.01 and gradient clip at norm 1. These hyper-parameters were chosen as standard defaults without hyper-parameter tuning. For the immediate-answer runs, we use a weight decay of 0.1 and gradient clip at norm 0.5; this change was made because using the original set of hyper-parameters leads to loss spikes and training instability.

For 2SUM training, we use a binary cross-entropy loss, since both inputs and chain-of-thought data are multi-hot vectors. Causal masking is applied as per the standard language-modelling objective.

## B  Probing Experiment

In this experiment, we validate that the effects of filler tokens on training are indeed caused by a difference in expressive capacity and not a difference in learning dynamics. To do so, we probe final-layer representations to confirm that filler-tokens are used for their additional computational capacity. This hypothesis is confirmed in Figure 5 where the first half of the filler tokens appear crucial, achieving 98% performance while using only 60% of the total filler tokens. Here each point represents a different final-layer fine tune varying the total filler-tokens available as a hyper-parameter. This early convergence given half the total filler tokens is to be expected for an efficient algorithm solving 3SUM, since each pair of inputs

---

[12]For 3SUM, $d_{input} \sim 10d + n$ following the notation of Equation (1). These dimensions correspond to tuple digits and hard-coded positional values.

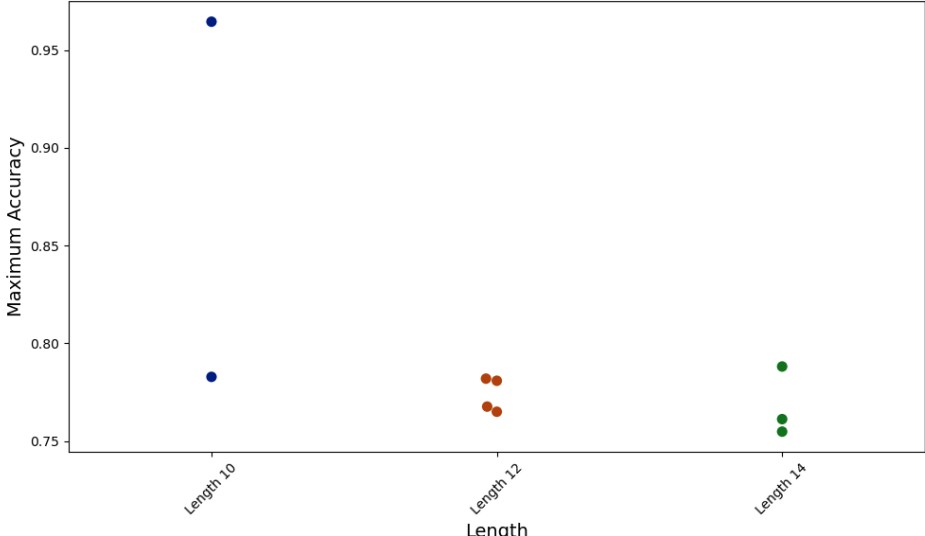

Figure 6: When training on mixtures of filler-token and instance-adaptive sequences, filler token sequences remain at baseline performance for lengths 12 and 14. For the length-10 seed showing $> 95\%$ accuracy, we found this model achieved 93.0% accuracy given only 10% of the original filler tokens, suggesting that the model makes minimal use of filler tokens. Points in this plot correspond to different random initializations.

needs to be summed only once for a total of $N^2/2$ comparisons–whereas the total number of filler tokens we provide is $N^2$.

Given the possibility of non-linear, learned probes confounding the interpretation of representations with the probes' own computation, we compare to the following control condition (Hewitt & Liang, 2019). This ensures that the observed filler-token vs accuracy scaling (Figure 5) reflects the frozen model layers' representations and *not* the probe itself. For the control task, we take a model trained with filler tokens on sufficiently simple 3SUM sequences for which immediate, no-filler, solutions are tractable: length-10, dimension-1 data.[13] To confirm that the probe results in Figure 5 reflect filler-token utility, we must confirm that the baseline probe on the dimension-1 control data does *not* find filler-token representations to be useful. As expected, we find that this length-10, dimension-1 model can achieve 100% accuracy given only 2% of the original number of filler tokens.[14] In effect, our probe finds filler token representations are redundant in models which have the expressive capacity to solve problems without filler tokens.

## C   Instance-Adaptive Chain of Thought

These chains of thought differ from parallelizable CoT in that they require caching sub-problem solutions in token outputs. When the computation carried out in a later chain-of-thought token depends on the results found in an earlier chain-of-thought token, we term this instance-adaptive computation. In these 3SUM chains of thought, the 3SUM problem is decomposed into dimension-wise 3SUMs: for each triple, a given dimension-wise summation is only computed if the previous dimension summed to zero–this is an instance-adaptive dependency. Instance-adaptive computation is incompatible with the parallel structure of filler token computation.

---

[13]This was determined by training another model without any intermediate tokens and observing that model achieved 100% accuracy.

[14]2% was the minimum number tested, it is likely no filler tokens are necessary.

### C.1 Data Generation Details

Inputs are drawn identically in the parallelizable and serial cases. The chain-of-thought generating process is as follows: Given input e.g. "A15 B75 C22 D13", the chain of thought is[15] ": A B C 15 75 22 2 B C D 75 22 13 0 ANS True". For each triple e.g. "A B C" and "B C D", the generating process is as follows:

1. List individual triple if it sums to zero in the first coordinate (e.g. the ': A B C' substring).

2. List the values of the triples, copying from the input (e.g. the '15 75 22' substring).

3. List the result of summing the given triple in the dimensions (e.g. the '2' substring, since $2 = (15 + 75 + 22 \bmod 10)_2$).

In our example, 'A B C' sum to 0 in the first dimension, but sum to 2 in the second dimension; 'B C D' sums to 0 in both dimensions meaning 3SUM is satisfied for this input.

### C.2 Results

Figure 6 shows that in 8/9 random intializations, instance-adaptive training fails to transfer to filler-token sequences. For these runs we use identical hyper-parameters to the parallel CoT setting, except we increase the number of epochs to 10 (this choice was arbitrary). In all settings, tuple dimensionality was fixed at 3.

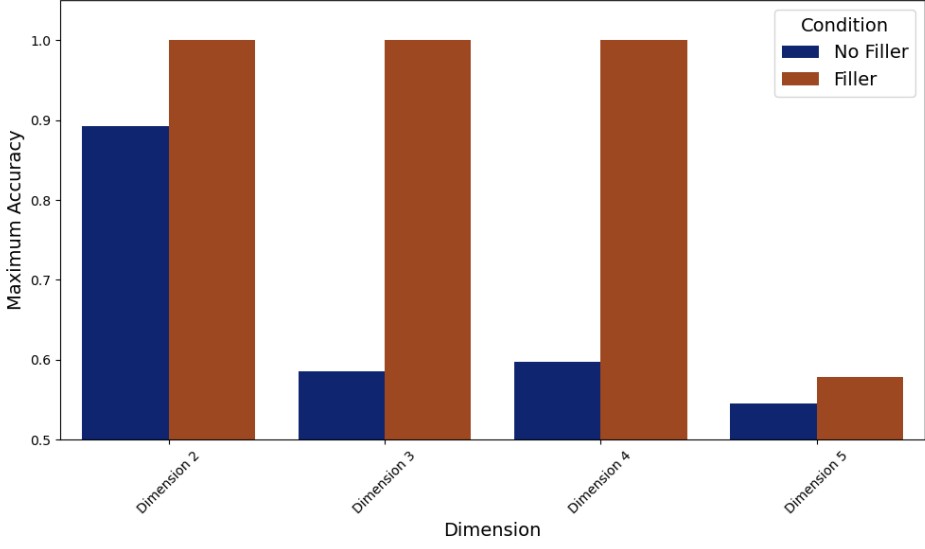

Figure 7: Performance on dimension-varied, length-10 data as a function of tuple dimensionality. The immediate-answer models were trained on a 50/50 mixtures of instance-adaptive CoT and immediate-answer sequences.

---

[15]In practice, as in the parallel case, we randomly drop dimensions to reduce sequence length, e.g. the post-drop sequence might be ": A B C $1_0 7_0 2_0$ 2 B C D $7_0 2_0 1_0$ 0 ANS True" if only the dimension 0 coordinates were kept.

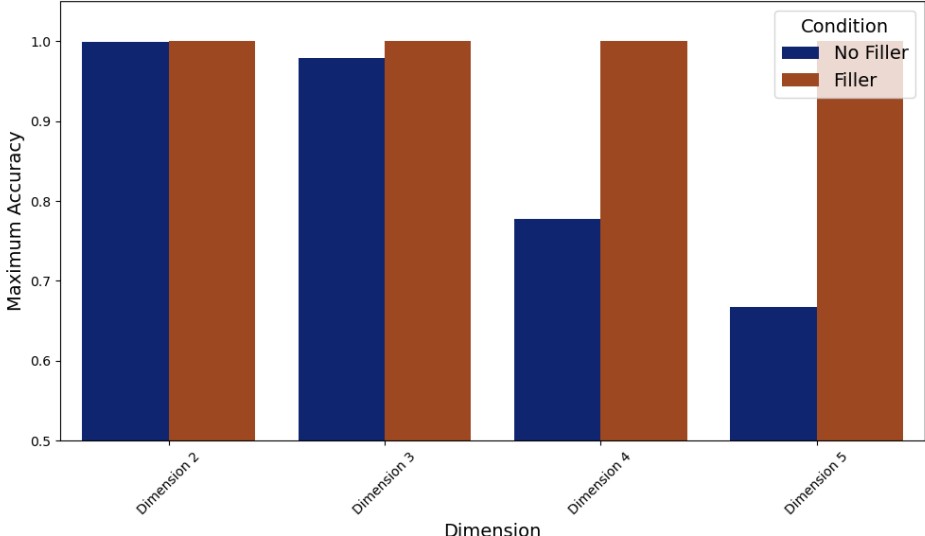

Figure 8: Performance on dimension-varied, length-8 data as a function of tuple dimensionality. These immediate-answer models were trained on no-intermediate-token sequences only. Note that these results under-perform relative to the mixed immediate-answer data models shown in Figure 4.

## D   Dimension Scaling Further Results

In Section 4.2, we saw that for length-8 inputs, scaling 3SUM dimensionality resulted in a performance gap between the filler-token and immediate-answer settings. In Figure 7, we show that this gap occurs at length-10 as well, but the emergence occurs at lower dimension: three-dimensional inputs show a filler-token performance gap, whereas six were required for Figure 4. Intuitively it is clear that 3SUM input length and dimensionality both contribute to the parameters required to solve problem instances in a single forward pass. Hence, the decrease in dimensionality required to realize a performance gap as we increase length.

For completeness, we also include a subset of length-8 results when using models trained only on immediate-response sequences. Unlike the other dimension-performance plots, the immediate-answer models trained for Figure 8 do not see any instance-adaptive chain-of-thought examples.

## E   Validation Loss Curves

We plot the validation loss curves for final-token, 'True' or 'False', prediction. Figure 9 shows that five epochs suffice for the most complex data. Figure 10 shows that 25 epochs are sufficient for evaluating immediate-answer performance in most cases. It is likely that the accuracy on the length-8 data shown in Figure 2 underestimates the unbounded-compute, limiting performance in this particular case. Given compute constraints, and the fact that none of our claims depend on the performance of this particular length-8 case, we did not explore this further.

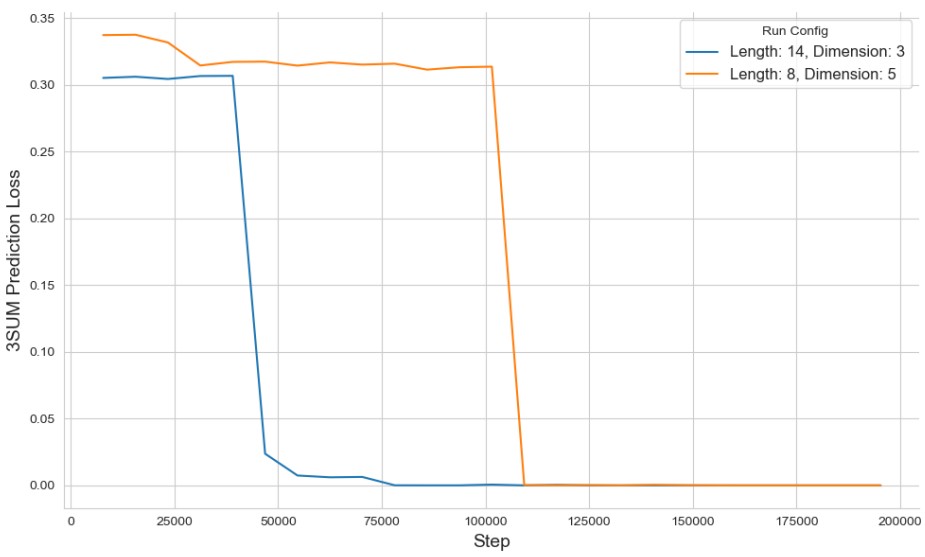

Figure 9: Validation loss on 3SUM final prediction token for models trained with filler. These models were trained for 5 epochs.

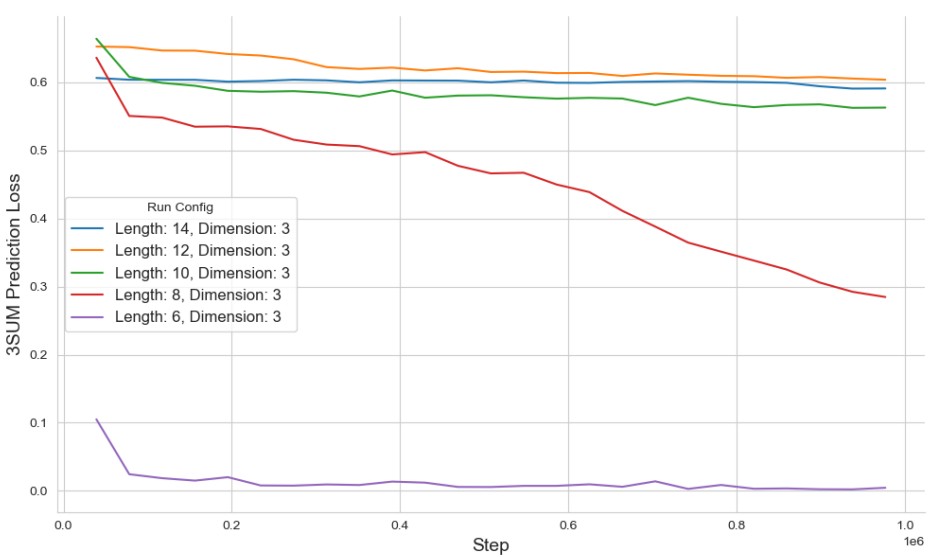

Figure 10: Validation loss on 3SUM final prediction token for models trained on no-intermediate-tokens, immediate-answer sequences. These models were trained for 25 epochs.

