# OpenReview forum: "Let’s Think Dot by Dot: Hidden computation in transformer language models"
_colmweb.org/COLM/2024/Conference — COLM_

### Official Review · Reviewer_f5kZ · 2024-04-21

**Rating:** 6
**Confidence:** 3
**Ethics Flag:** 1

**Summary:**

This paper proposes two experiments to show that transformers benefit from longer sequences to make implicit computations for tasks that require stepwise computations where intermediate results depend on previous results. The main experiment is the 3-sum problem, where models need to find if there exist three elements in the list that sum to 0 mod ten on all dimensions. The authors create a synthetic dataset that supervises this task with meaningful CoT sequences and filler non-meaningful sequences. Using a model trained on this data, the authors show that the model improves significantly with non-meaningful filler tokens compared to no filter tokens. They use this to suggest that models benefit from filler tokens to perform implicit computations for better reasoning.

**Reasons To Accept:**

The overall experiment is interesting and provides empirical evidence on whether intermediate, meaningful, or tokens can be beneficial for the models to perform implicit sequential computations.

**Reasons To Reject:**

I am not fully convinced by the main conclusions presented in this paper.

1. The authors claim to draw a conclusion between TC^0 complexity class and the need for filler tokens as intermediate computations. However, I do not see how the 3-sum and 2-sum-transform experiments are connected with this complexity class in general and why this conclusion can be generalized based on these two experiments.

2. The authors claim that models use filler tokens to perform more computation, but if I am not mistaken, this conclusion is directly suggested by how the experiment is set up, so I am not sure what point the authors are trying to make on top of it. With the filler tokens during the training stage, the model literally performs more computation based on the representation of the filler tokens and the positional embeddings; more computation usually equals more overfitting to the task and sometimes better performances. Especially given the actual CoT sequences in the training stage, the model may learn to generalize based on positional embeddings of how the actual computation should be, hence making the filler tokens not "non-meaningful" anymore. I suggest the authors remove the meaningful CoT tokens in training, freeze the filler word representations, and experiment with #layers v. fillers.

3. The authors argue that transformers do not produce human-like, faithful serial reasoning, as the experiments suggest. I think this claim is overall true, but the experiments cannot derive this logically. The proposed experiments show that the transformers CAN benefit from non-interpretable implicit computations, but they do not show that they CANNOT achieve the same performance with human-like reasoning.

Overall, I do not understand the authors' main points, but I am happy to revise if they point out my misunderstandings.

---

> ### Author Rebuttal · Authors · 2024-05-31
>
> Thank you for engaging with our work!
>
> #1, How 3SUM relates to TC0: TC0 can be equivalently defined as first-order logic with majority quantifiers. In Figure 3, we show how 3SUM can be naturally defined in first-order logic and is thus in TC0. Crucially, it also requires three quantifiers, which we propose as the key feature for a TC0 problem that transformers without filler would not be able to express. This is why we are interested in 3SUM. We will add some substantial additional discussion to clarify.
>
> #2 “The authors claim that models use filler tokens to perform more computation, but … this conclusion is directly suggested by how the experiment is set up” To clarify what is and what is not immediate from the experimental setup, we emphasize that there are two separate questions addressed by our work:
>
>     (1) Theoretically/Expressivity: Do transformer weights exist which can use filler tokens to solve problems unsolvable via immediate answer? Yes as described above in #1.
>
>     (2) Empirically/Learning: when can a transformer learn via SGD to use filler tokens? Theory alone does not tell us whether this solution can be efficiently learned via SGD. This is where our experiments come in, showing empirically that filler tokens pose a hard learning problem (sec 4.3).
>
> #3 We agree with you that “transformers CAN benefit from non-interpretable implicit computations, but they do not show that they CANNOT achieve the same performance with human-like reasoning.”
> We agree that CoT is generally useful for problems beyond TC0 whereas filler is only useful for a subset of TC0. This sentence from the intro reinforces your point:
> > “Whereas linear or polynomial chain of thought steps can add power … beyond TC0, transformers remain in TC0 with [filler tokens] … Filler tokens might extend the expressive power … within TC0.”
>
> So the performance hierarchy is:	immediate-answer < filler tokens << chain of thought
>
> For 3SUM, in particular, we mention in section 4.2 that human-like CoT achieves perfect performance: “Models converge to 100% accuracy on chain-of-thought data as well, but we do not show this in figures for simplicity”. This is an important point, and so to improve clarity we will move this from footnote 9 to main-text.

---

> > ### Comment · Reviewer_f5kZ · 2024-06-05
> > **Thanks for the rebuttal**
> >
> > Thanks for providing the rebuttal.
> >
> > Let me clarify my first point: I understand that 3-sum and 2-sum are TC0 problems, but I do not understand how the conclusions derived from experiments on these two problems can be generalized to the TC0 class in general. The authors seem to be drawing too strong a conclusion based on limited empirical evidence, which I suggest the authors clarify in the final version.
> >
> > Let me clarify my second point: I suggested that the models only learn meaningful representations for the filler tokens because the training data contains meaningful tokens (CoTs), so the models can assign a meaningful representation based on actual CoT semantics through positional relations. For example, if the model sees both "the weather is good" and "the weather is .", the last dot could be semantically related to "weather." Could you please clarify if this could be the case? If this happens, the overall contribution will be less interesting because the model relies more on positional embeddings and performs CoT implicitly in the filler-token settings.
> >
> > I am happy with the last point and willing to change my score if the previous question can be answered.

---

> > > ### Author Response · Authors · 2024-06-05
> > >
> > > Thank you for this update on your questions!
> > > 1. We do not claim that 3SUM is representative of all of TC0 (more on this in comments [above](https://openreview.net/forum?id=NikbrdtYvG&noteId=YBOYaSGiZl)), but rather that 3SUM is representative of quantifier resolution problems having quantifier depth >2. Starting from our theoretically-motivated prediction that filler-tokens provide an expressive advantage for such quantifier resolution problems, we present filler-token success on 3SUM as empirical evidence (analogous to an existence proof) of the relevance of filler to problems of quantifier depth >2. These results do not suffice to show that all such quantifier resolution problems are solvable using filler; we will update language used in our paper to precisely reflect what our results show.
> > > 2. Regarding semantics, while we agree that the learned semantics of CoT in general are an important object to study, in our case (2SUM and 3SUM) the semantics are not a useful lens, since we are studying an algorithmic problem in which memorization/dataset-wide associations are irrelevant. In the algorithmic setting, the distinguishing feature between CoT and filler tokens is that CoT involves serial computation (increasing depth of circuit) while filler involves parallel computation (increasing width of circuit). On this view, the “parallelizable CoT solution” we train models on does, in principle, allow circuit reuse where the model executes similar activation patterns across both CoT and filler-token sequences. On the other hand, when training on “serial CoT solution” different circuits are required for CoT versus filler-token sequences. So indeed, we find that on 3SUM, models only succeed in learning to use filler tokens when the circuits learned on CoT match those needed for filler tokens, with the serial CoT-trained models failing to learn filler tokens. We agree that these results suggest filler-token use is limited since we expect most natural language CoT instances to involve serial computation. However, we see this as a contribution of our work rather than a drawback: to understand the broader prevalence of filler-token use, it is important to understand under what conditions filler-token use can be learned. Our results provide insight on this question showing that empirically, learning filler-token solutions is significantly harder in the absence of relevant, parallelizable CoT supervision for 3SUM.

---

> > > > ### Comment · Reviewer_f5kZ · 2024-06-05
> > > >
> > > > Thanks for the answers, which are fair points to me. I suggest the authors verify all claims in the paper, make the language precise, and clarify the second point. With such promises, I increase the overall score.

---

### Official Review · Reviewer_7L8q · 2024-05-05

**Rating:** 4
**Confidence:** 4
**Ethics Flag:** 1

**Summary:**

This work studies how seemingly meaningless filler tokens can be used as a kind of zero-shot chain-of-thought to solve algorithmic problems. For example, it can empirically identify some classes of problems where filter tokens are useful.

**Questions To Authors:**

Please refer to the Weakness above.

**Reasons To Accept:**

1. This work addresses an interesting problem of how filler tokens can be used as a useful chain-of-thought method.

2. This work makes several interesting empirical observations on the filler tokens as CoT, as highlighted with bold text or titles in section 4.2-4.3.

**Reasons To Reject:**

My major impression toward this work is that although it tackles an interesting problem, it is still largely preliminary and requires more in-depth study to fully materialize its key idea. This submission could be acceptable to a relevant workshop but is still below the bar of a conference paper.

1. 3SUM and 2SUM are interesting problem but still preliminary. I believe more sets of problems should be explored to verify the effects of filler tokens as CoT.

2. Experiments should be improved to be more thorough in multiple dimensions.
- More variants of 3SUM problems may be tested with different lengths, dimension, etc.
- More baselines of CoT may need to be compared.

---

> ### Author Rebuttal · Authors · 2024-05-31
>
> Thank you for taking time to engage with our work. Our goal in this paper was to construct an empirical and theoretical existence proof demonstrating conditions under which filler tokens can be useful for transformer LMs. Past work has shown that, for general tasks, LMs fail to benefit from filler tokens. Our contribution is to show that, while filler tokens do not provide a universal computational benefit, they do help on specific algorithmic problems involving parallel search (specifically, first-order logic problems having quantifier depth >2). Our experiments with 3SUM confirm this theoretically motivated hypothesis on a specific instance of such a problem. Our experiments establish that, across data complexity, sequence length and control conditions, filler tokens provide performance improvements on algorithmic problems involving parallel search. This is a significant contribution since previous work had only ever shown LLMs failing to benefit from filler tokens ([1](https://arxiv.org/abs/2307.13702)), whereas our results establish a specific case where they can help.
>
> Regarding further baselines, we note that our CoT setting achieves 100% accuracy, and so “More baselines of CoT” would not make a difference. Regarding running additional experiments varying 3SUM dimension. Indeed, we have run experiments up through 7 dimensional inputs and have found the results are qualitatively similar. We will include the extended figure in the final version.
>
> > “I believe more sets of problems should be explored to verify the effects of filler tokens as CoT"
>
> In contrast to broad benchmarking work as in e.g. [BigBench](https://github.com/google/BIG-bench), we aim to focus on specific tasks where theory suggests filler tokens would be useful rather than exhaustively benchmarking LMs across many different, less motivated, tasks.

---

### Official Review · Reviewer_KfHS · 2024-05-11

**Rating:** 7
**Confidence:** 4
**Ethics Flag:** 1

**Summary:**

This paper discusses whether adding dummy tokens for LLMs can enhance their expressive power. The authors design two synthetic datasets for the evaluation. Results show that even if adding dummy tokens the circuit complexity class remains TC0, but extra power is indeed added.

**Reasons To Accept:**

1. This paper explores a very important problem, which provides insights into understanding how chain-of-thought (CoT) works for LLMs.
2. The synthetic tasks are well-designed. Results are quite interesting, and can support the proposed claim pretty well.

**Reasons To Reject:**

I’m afraid the gap between practical LLMs on real data and the toy model on synthetic data cannot be ignored. The pretraining corpus and pattern are quite simple and clean. However, in practice, the LLM is pretrained on massive data with various hidden structures. The analysis conclusions as well as hidden assumptions may not hold in practice. However, considering that it’s extremely difficult to implement analysis for real LLMs. I would still think this is a pretty good work.

---

> ### Author Rebuttal · Authors · 2024-05-31
>
> Thank you for your review! We're glad you found the experiments compelling. We agree that investigating filler tokens with large-scale LMs is an interesting question, although we see it as outside the scope of the current project which is focused on analyzing the benefit of filler tokens in a targeted setting where we expect them to be useful.

---

### Official Review · Reviewer_zMzE · 2024-05-14

**Rating:** 6
**Confidence:** 3
**Ethics Flag:** 1

**Summary:**

This paper investigates the role of filler tokens in transformer language models, particularly in enhancing computational capabilities independently of meaningful chain-of-thought tokens. The authors show that transformers can solve complex algorithmic tasks using meaningless filler tokens, "......" in this work, and provide empirical and theoretical insights into the conditions where these tokens are beneficial.
They build new synthetic datasets for nested quantifier resolution and training on LLAMA-34M from scratch. Compared with vanilla inference, it demonstrates that filler tokens can enable hidden computation and improve results, especially when scaling the length and dimension of 2 and 3Sum tasks.

**Questions To Authors:**

1. How do you anticipate the findings would scale with much larger pre-trained transformer models?
2. In Table 1, does chain-of-thought include both parallelizable and serial tokens, or only parallelizable tokens?
3. In Table 1, if the "No Intermediate Tokens" is the same as no-filler, refer to it as no-filler to avoid misunderstanding.
4. Section 4.3 is quite ambiguous; the authors should provide detailed experimental tables and prove their claims for a better explanation.
5. In the example  “A05 B75 C22 D13 : AB 70 AC 27 AD 18 BC 97 BD 88 CD B ANS True”,  shouldn't the "AB 70" be "AB 80"?

**Reasons To Accept:**

1. The paper provides a unique perspective on how filler tokens, often considered meaningless, can enhance the computational capabilities of transformer models.
2. It combines theoretical characterizations with empirical evidence, offering a robust examination of the utility of filler tokens.
3. The findings have provided valuable insights for designing and interpreting chain-of-thought reasoning in large language models, potentially impacting future model training and evaluation strategies.

**Reasons To Reject:**

1. The authors note that learning to use filler tokens requires dense, task-specific supervision, which may not be practical or scalable in many contexts.
2. The experiments are conducted on relatively small transformer models, which may not fully capture the behavior and potential of larger, more advanced models
3. The naming of different methods is messy and needs to be unified (Filler, No filler, No Intermediate Tokens, CoT...), especially in Table 1.

---

> ### Author Rebuttal · Authors · 2024-05-31
>
> Thank you for the interest in our paper! Glad you found the experimental results and theoretical framing on filler tokens robust and valuable! To frame this discussion, our goal with this paper was to identify and explain conditions under which models can learn to use filler tokens, since previous work had only ever shown LLMs failing to use filler tokens ([1](https://arxiv.org/abs/2307.13702),[2](https://www.lesswrong.com/posts/oSZ2xTxEMZh9f3Yaz/llms-are-mostly-not-helped-by-filler-tokens)).
>
> To go through point-by-point:
>
> - “learning to use filler tokens requires dense, task-specific supervision which may not be practical or scalable in many contexts.”
> Our paper aims to advance the science and understanding of LMs and it’s not a methods paper. We set out to identify criteria under which filler tokens can benefit model performance and this is one such criterion (filler requires dense, task-specific supervision). So, the fact that learning to use filler appears difficult is an insight provided by our work, and we do not see it as a drawback.
> - On Table 1, the three settings are best described as “Filler tokens”, “Chain of thought” and “Immediate answer”. “Immediate answer” is synonymous with “no intermediate tokens” and “no filler tokens”, to make this point clear we will update to use “immediate answer” throughout.
> - “AB70” is correct since (0,5)+(7,5)%10=(7,0).
> - To clarify the serial CoT controls, we will include an additional appendix figure showing detailed results for sec 4.3.
> - Regarding larger pre-trained transformers, we expect that out-of-place computation will/has emerged, but given our results around the difficulty of learning to use filler tokens such cross-token hidden computation appears less likely absent targeted training.

---

> > ### Comment · Reviewer_zMzE · 2024-06-06
> >
> > Thank you for your rebuttal. I appreciate its theoretical value and eagerly anticipate its potential real-world application.
> > Unifying the naming conventions and clarifying the serial CoT controls should enhance the reader's understanding of the paper.
> > After considering all the rebuttals, I will keep my score at 6.

---

### Comment · Area_Chair_iFxu · 2024-05-31
**One More Question from AC**

To the best of my knowledge, there are several **consistent theoretical investigations** regarding the expressiveness of Transformer models, including both encoder setting and autoregressive decoder setting:

[1]. The parallelism tradeoff: Limitations of log-precision transformers

[2]. Transformers learn shortcuts to automata

[3]. Towards Revealing the Mystery Behind Chain of Thought: A Theoretical Perspective

[4]. The Expressive Power of Transformers with Chain of Thought

All these theoretical findings collectively contribute to understanding the representational capabilities and limitations of Transformers. They suggest that while a Transformer encoder may never surpass TC0 complexity, an autoregressive model equipped with Chain of Thought (CoT) exhibits significantly greater power. However, your empirical observations **appear to contradict** certain aspects of these theoretical assertions. Could you explain this problem? And what is your new insight in this direction compared to these already-known results?

Thanks,

---

> ### Author Response · Authors · 2024-05-31
>
> Thanks for your question! There is no contradiction between those past works and our work. In fact, our results are motivated by and build on the papers you mention, but we investigate filler tokens (blank tokens get added to the input) as opposed to traditional CoT addressed in prior work (where model-generated tokens get added to the input).
>
> [1] shows that transformers, without CoT, can express at most the class TC0. With CoT, i.e. the transformer can generate tokens that get appended to the input, power extends beyond TC0 [3, 4]. As we discuss in Section 2, this is *not* the case for transformers with filler tokens, which remain in TC0 by the argument from [1]. Applied to filler tokens, [1] implies the following proposition:
> > *Transformers with filler tokens remain in TC0.* Proof sketch: For inputs of size n, a transformer can be simulated by a constant depth, poly(n) size threshold circuit (TC). So, if we add polynomial filler tokens, we can still simulate the transformer with a constant depth and poly(poly(n)) = poly(n) size circuit.
>
> Thus, filler tokens will not increase expressive power as much as CoT.
>
> Our argument is that there are likely functions in TC0 that transformers without filler tokens cannot express but which transformers can likely express with filler tokens. Namely, these functions are TC0 functions with >2 quantifier depth, such as 3SUM, which we consider in our experiments. Our experimental results that transformers with filler tokens can express this function are thus consistent with prior work and our hypothesis because a) 3SUM is in TC0 and b) it requires >2 quantifiers to express, so we expect filler tokens to be necessary to express it.
>
> One of the high-level takeaways from this work is that CoT and filler tokens extend the power of transformers in different ways. CoT adds depth to the computation graph, allowing transformers to express functions *outside* TC0. On the other hand, filler tokens extend the width (or quantifier depth) of the computation graph, likely allowing larger parallel computation *within* TC0. We will make this clearer in revisions.
>
> The empirical evidence we find complements theory in two ways (1) Our work empirically demonstrates functions in TC0 that transformers without filler tokens fail to learn, but which transformers can learn with filler tokens. These empirical results provide evidence of the above-mentioned hypothesis that there is a finer-grained expressivity distinction within TC0 between transformers with filler tokens and transformers without filler tokens. (2) Our work also complements the prior theoretical work you cite, as expressivity theory results alone *do not* tell us whether the theoretical transformer constructions used in expressivity proofs can also be efficiently learned via SGD—learnability is a stronger criterion than expressivity. Our experiments on 3SUM demonstrate that without our particular training data (additional, parallelizable CoT sequences) learning to use filler tokens poses an intractable learning problem, see sec 4.3.

---

> > ### Comment · Area_Chair_iFxu · 2024-06-01
> >
> > Thanks for the response.
> >
> > "There are likely functions in TC0 that transformers without filler tokens cannot express but which transformers can likely express with filler tokens."——Is this a rigorous statement or just a hypothesis that you try to demonstrate empirically through experiments?
> >
> > AC

---

> > > ### Author Response · Authors · 2024-06-03
> > >
> > > The first part of the quoted claim is that transformers cannot express all problems in TC0. This has not been proven but follows from the conjecture that there is a size hierarchy within TC0 (see Conclusion from [Merrill et al., 2023](https://arxiv.org/abs/2210.02671)). The existence of a size hierarchy (meaning, larger TC0 circuits are more expressive than smaller ones) has not been proven to our knowledge, but it is a quite intuitive conjecture, and [recent work in complexity theory](https://arxiv.org/abs/1809.04092) has established size hierarchy theorems for circuit classes related to TC0 (but not TC0 itself).
> > >
> > > The second part of the cited claim is that transformers with filler tokens likely achieve greater expressive power within TC0. This is again related to the size hierarchy conjecture, because one way to understand the benefit of filler tokens is they increase the *size* of the circuit or computation graph for the transformer (while keeping the *depth* fixed). Thus, if larger TC0 circuits are more expressive than smaller ones (meaning there is a size hierarchy), we should expect filler tokens could increase expressive power.
> > >
> > > A type of problems where circuit size seems particularly helpful is problems with nested quantifiers, because larger circuits allow expressing quantification over many variables at once. We will try to update the motivation for 3SUM and discussion for quantifier depth to make the connection to circuits more clear.

---

> > > > ### Comment · Area_Chair_iFxu · 2024-06-04
> > > >
> > > > Thanks for the input. I will work together with reviewers and take these responses into consideration.
> > > >
> > > > AC

---

### Decision · Program_Chairs · 2024-07-10

**Decision:**

Accept

**Comment:**

This paper explores the impact of filler tokens in transformer language models, particularly in improving computational capabilities independently of meaningful chain-of-thought (CoT) tokens. The authors present empirical and theoretical insights into how these tokens can benefit transformer models in solving complex algorithmic tasks. Key findings demonstrate that filler tokens can enable hidden computation, improving performance on tasks like the 2Sum and 3Sum problems as sequence length and dimensionality scale.

The authors had adequate discussions with AC and the reviewers. Although the general claim has not been rigorously proved, the empirical findings are interesting. Addressing the noted weaknesses and expanding the scope of experiments would significantly enhance its contributions to the field.

[comments from the PCs] Please follow up on the AC and reviewer comments to refine the claims of the paper and experiments.